# SARS-CoV-2 viral dynamics in non-human primates

**Antonio Gonçalves**[1]*, **Pauline Maisonnasse**[2], **Flora Donati**[3,4], **Mélanie Albert**[3,4], **Sylvie Behillil**[3,4], **Vanessa Contreras**[2], **Thibaut Naninck**[2], **Romain Marlin**[2], **Caroline Solas**[5], **Andres Pizzorno**[6], **Julien Lemaitre**[2], **Nidhal Kahlaoui**[2], **Olivier Terrier**[6], **Raphael Ho Tsong Fang**[2], **Vincent Enouf**[3,4,7], **Nathalie Dereuddre-Bosquet**[2], **Angela Brisebarre**[3,4], **Franck Touret**[8], **Catherine Chapon**[2], **Bruno Hoen**[9], **Bruno Lina**[6,10], **Manuel Rosa Calatrava**[6], **Xavier de Lamballerie**[8], **France Mentré**[1], **Roger Le Grand**[2], **Sylvie van der Werf**[3,4], **Jérémie Guedj**[1]

1 Université de Paris, IAME, INSERM, Paris, France, 2 Université Paris-Saclay, Inserm, CEA, Center for Immunology of Viral, Auto-immune, Hematological and Bacterial diseases (IMVA-HB/IDMIT), Fontenay-aux-Roses & Le Kremlin-Bicêtre, France, 3 Unité de Génétique Moléculaire des Virus à ARN, GMVR: Institut Pasteur, UMR CNRS 3569, Université de Paris, Paris, France, 4 Centre National de Référence des Virus des infections respiratoires (dont la grippe), Institut Pasteur, Paris, France, 5 Aix-Marseille Univ, APHM, Unité des Virus Emergents (UVE) IRD 190, INSERM 1207, Laboratoire de Pharmacocinétique et Toxicologie, Hôpital La Timone, Marseille, France, 6 CIRI, Centre International de Recherche en Infectiologie, (Team VirPath), Univ Lyon, Inserm, U1111, Université Claude Bernard Lyon 1, CNRS, UMR5308, ENS de Lyon, Lyon, France, 7 Plate-forme de microbiologie mutualisée (P2M), Pasteur International Bioresources Network (PIBnet), Institut Pasteur, Paris, France, 8 Unité des Virus Emergents, UVE: Aix Marseille Univ, IRD 190, INSERM 1207, IHU Méditerranée Infection, Marseille, France, 9 Emerging Diseases Epidemiology Unit, Institut Pasteur, Paris, France, 10 Laboratoire de Virologie, Centre National de Référence des Virus des infections respiratoires (dont la grippe), Institut des Agents Infectieux, Groupement Hospitalier Nord, Hospices Civils de Lyon, Lyon, France

* antonio.goncalves@inserm.fr

**Data Availability Statement:** All relevant data are within the manuscript and its Supporting Information files.

**Funding:** This work was funded by the French national research agency (ANR) through the

## Abstract

Non-human primates infected with SARS-CoV-2 exhibit mild clinical signs. Here we used a mathematical model to characterize in detail the viral dynamics in 31 cynomolgus macaques for which nasopharyngeal and tracheal viral load were frequently assessed. We identified that infected cells had a large burst size (>10^4 virus) and a within-host reproductive basic number of approximately 6 and 4 in nasopharyngeal and tracheal compartment, respectively. After peak viral load, infected cells were rapidly lost with a half-life of 9 hours, with no significant association between cytokine elevation and clearance, leading to a median time to viral clearance of 10 days, consistent with observations in mild human infections. Given these parameter estimates, we predict that a prophylactic treatment blocking 90% of viral production or viral infection could prevent viral growth. In conclusion, our results provide estimates of SARS-CoV-2 viral kinetic parameters in an experimental model of mild infection and they provide means to assess the efficacy of future antiviral treatments.

## Author summary

Non-human primates infected with SARS-CoV-2 develop a mild infection resembling asymptomatic human infection. Here we used viral dynamic modelling to characterize

TheraCoV ANR-20-COVI-0018 (JG) and also by the Bill & Melinda Gates Foundation through INV-017335 (JG). The funders had no role in study design, data collection and analysis, decision to publish, or preparation of the manuscript.

**Competing interests:** I have read the journal's policy and the authors of this manuscript have the following competing interests: A.G. PhD grant has been provided by ROCHE Company. J.G. has worked as consultant for ROCHE Company.

the nasopharyngeal and tracheal viral loads. We found that viral load rapidly declined after peak viral load despite the absence of association between model parameters and immune markers. The within-host reproductive basic reproduction number was approximately equal to 6 and 4 in nasopharynx and trachea suggesting that a prophylactic therapy blocking viral entry or production with 90% efficacy could be sufficient to prevent viral growth and peak viral load.

## Introduction

The severe acute respiratory syndrome coronavirus 2 (SARS-CoV-2) which originated in Wuhan, China, at the end of December 2019, has spread rapidly around the world, resulting at the end of December 2020 in more than 1,600,000 deaths [1]. Fortunately, the majority of infections do not lead to hospitalizations [2], and the vast majority of subjects infected with SARS-CoV-2 will experience asymptomatic or pauci-symptomatic infection characterized by specific (anosmia) or general symptoms (fever, fatigue) [3–5]. In other acute or chronic viral diseases (HIV, HCV, influenza), the characterization of viral load kinetics has played an important role to understand the pathogenesis of the virus and design better antiviral drugs [6–8]. In the case of SARS-CoV-2, viral kinetics has been found to be associated with mortality in hospitalized patients [9] but the association in less severe patients is unclear. This is due to the fact that many studies rely on large cross-sectional analyses with few patients having serial data points or, in contrary, on detailed small series of patients [10–12]. In that perspective, the analysis of data generated in non-human primates is a unique opportunity to characterize in detail the viral dynamics during natural infection, and to study the effects of antiviral therapy [13–15].

Here, we used data generated on nonhuman primates (NHP) and treated with hydroxychloroquine (HCQ) ± azithromycine (AZTH) in either pre- or post-exposure prophylaxis [16] to develop a mathematical model of SARS-CoV-2 infection. Although our analysis did not reveal any significant antiviral efficacy of HCQ, the large data generated (31 cynomolgus macaques with frequent measurements of both nasopharyngeal and tracheal viral loads) allowed us to characterize in detail the key parameters driving the viral dynamics and to evaluate putative immune response mechanisms during the infection.

## Methods

### Ethics statement

Cynomolgus macaques aged 37–40 months and originating from Mauritian AAALAC certified breeding centers were used in this study. Animals were housed under BSL-2 and BSL-3 containment when necessary (Animal facility authorization #D92-032-02, Prefecture des Hauts de Seine, France) and in compliance with European Directive 2010/63/EU, the French regulations and the Standards for Human Care and Use of Laboratory Animals, of the Office for Laboratory Animal Welfare (OLAW, assurance number #A5826-01, US).

The protocols were approved by the institutional ethical committee "Comité d'Ethique en Expérimentation Animale du Commissariat à l'Energie Atomique et aux Energies Alternatives" (CEtEA #44) under statement number A20-011. The study was authorized by the "Research, Innovation and Education Ministry" under registration number APAFIS#24434-2020030216532863v1.

## Experimental procedure

Our study includes 31 cynomolgus macaques (16 male, 15 female) infected with $10^6$ pfu (corresponding to $10^{10}$ total RNA copies) of a primary isolate of SARS-CoV-2 (BetaCoV/France/IDF/0372/2020). Each animal received 5 mL of the total inoculum: 90% (4.5 mL) were injected by the intra-tracheal route and 10% (0.5 mL) by the intra-nasal route [17,18]. Nasopharyngeal and tracheal swabs were collected longitudinally at days 1, 2, 3, 4, 5, 7, 9, 11, 13, 16 and 20 post-infection (pi) and eluted in 3 mL of Universal transport medium (Copan) or Viral Transport Medium (CDC, DSR-052-01). Viral RNA levels were assessed in each sample using a real-time PCR, with 8540 and 180 copies/mL as quantification and detection limits, respectively (**Fig 1**)

The original study included 6 groups treated either by a high dosing regimen (Hi) of HCQ (90 mg/kg loading dose and 45 mg/kg maintenance dose) ± AZTH (36 mg/kg of at 1 dpi, followed by a daily maintenance dose of 18 mg/kg), a low dosing regimen (Lo) of HCQ (30 mg/kg loading dose and 15 mg/kg maintenance dose) or a vehicle (water) as a placebo. Among the 23 treated animals, 14 were treated at 1 dpi with a Lo (n = 4), Hi (n = 5) dose of HCQ or HCQ +AZTH (n = 5). A late treatment initiation was also investigated in 4 animals receiving a Lo dosing regimen. Finally, 5 animals received HCQ at the Hi dose starting at day 7 prior to infection as a pre-exposure prophylaxis (PrEP) (see **S1 Data**)

## Viral dynamic model of SARS-CoV-2

**Model describing nasopharyngeal and tracheal swabs.**   Given that HCQ was not associated to any antiviral effect [16], we pooled the data from either treated and untreated groups. Thus, we consider all animals as controls in our primary analysis but we also investigated a potential effect of HCQ (see details in **S1 Text**). Nasopharynx and trachea were considered as two distinct compartments of the upper respiratory tract (URT) where each one is described by a target cell limited model [13,19,20]. In this model, susceptible cells (T) are infected by an infectious virus ($V^I$) at a rate β and generate non-productive infected cells ($I_1$). After an eclipse

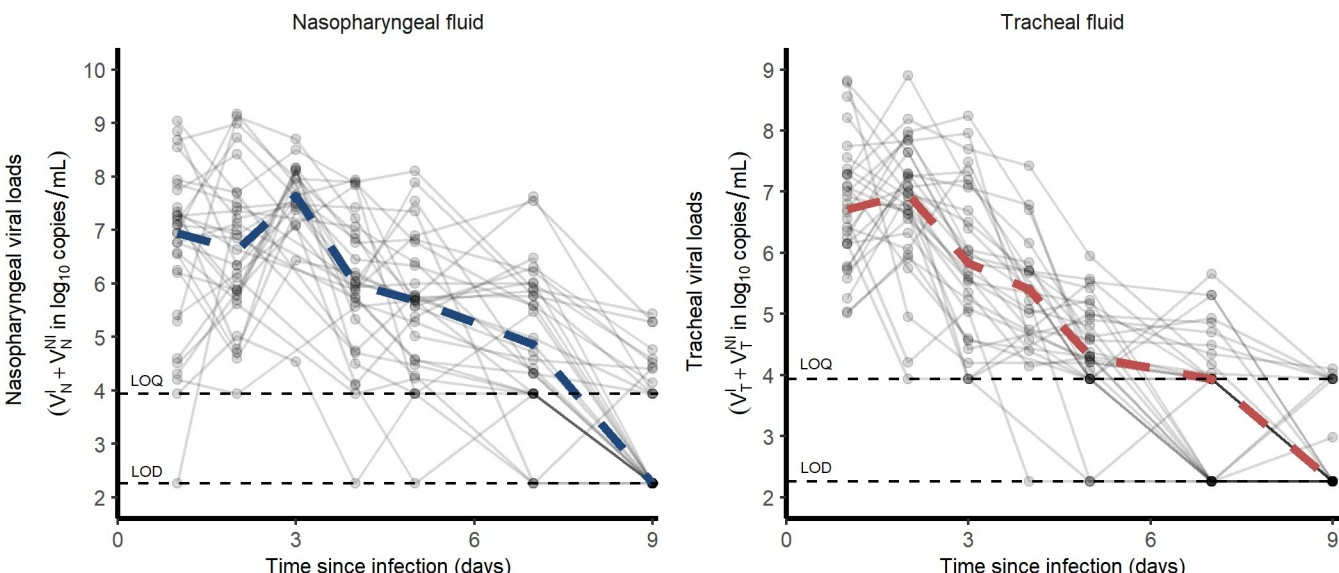

**Fig 1. Nasopharyngeal and tracheal SARS-CoV-2 viral loads in infected cynomolgus macaques treated by a placebo or hydroxychloroquine ± azithromycin.** Dashed lines represent the lower limit of detection (LOD) and lower limit of quantification (LOQ).

phase of duration $1/k$ on average, infected cells start to be productive ($I_2$) and produce both infectious and non infectious viruses at a rate $\mu p$ and $(1-\mu)p$ respectively. Productive infected cells die at a rate $\delta$ and viruses are cleared at a rate $c$ (see **S2 Text**). The model is given by ordinary differential Eqs (1) to (5) where subscript X denotes the compartment of interest either nasopharynx or trachea. The basic reproduction number is $R_0 = \frac{\beta p T_0 \mu}{c\delta}$ and the burst-size is $N = \frac{p}{\delta}$.

$$\frac{dT_X}{dt} = -\beta_X T_X V_X^I \tag{1}$$

$$\frac{dI_{1,X}}{dt} = \beta_X T_X V_X^I - kI_{1,X} \tag{2}$$

$$\frac{dI_{2,X}}{dt} = kI_{1,X} - \delta_X I_{2,X} \tag{3}$$

$$\frac{dV_X^I}{dt} = p_X I_{2,X}\mu - cV_X^I \tag{4}$$

$$\frac{dV_X^{NI}}{dt} = p_X I_{2,X}(1-\mu) - cV_X^{NI} \tag{5}$$

**Fixed parameters.** Because not all parameters can be estimated when only total viral RNA are measured, some parameters had to be fixed based on the experimental setting or the current literature. As only the product $p \times T_0$ is identifiable, we chose to fix $T_0$, the initial concentrations of susceptible cells, as follows. We measured the surfaces (S) and volumes (V) of the nasopharynx and the trachea in one euthanized animal and obtained $V_N = 6.3$ $mL$; $V_T = 1.2$ $mL$; $S_N = 50$ cm$^2$; $S_T = 9$ cm$^2$. Assuming an apical surface of one epithelial cell of $s = 4\ 10^{-7}$ cm$^2$/cell [21], the number of target cells exposed to the virus in the nasopharynx and the trachea are $\frac{S_N}{s \times V_N} = 1.98 \times 10^7$ $cells/mL$ and $\frac{S_T}{s \times V_T} = 1.88 \times 10^7$ $cells/mL$, respectively. Only a fraction of these cells expresses both angiotensin-converting enzyme 2 (ACE2) and the type II transmembrane serine protease (TMPRSS2), and we fixed this proportion to 0.1% i.e. $T_{N,0} = 1.98 \times 10^4$ and $T_{T,0} = 1.88 \times 10^4$ cells in nasopharynx and trachea, respectively. Also, we supposed that the eclipse phase duration was equal in the nasopharynx and in the trachea, and we set $k = 3$ d$^{-1}$ based on *in vitro* studies showing that virus release occurs 8 hours after infection [22]. Third, we supposed the proportion of infectious virus $\mu$ remained constant over time. TCID$_{50}$ is a proxy for the infectious viruses. **Fig 2B** shows that the ratio TCID$_{50}$/total viral loads is $10^{-5}$ (range $10^{-6}$–$10^{-4}$). However, not all the infectious viruses present in the samples may be detected in Vero-E6 cells experiments. Hence, we fixed $\mu$ to $10^{-4}$, the upper bound of the observed ratio of infectious virus. Finally the viral clearance $c$ was fixed to 10 d$^{-1}$, consistent with the rapid viral clearance rate of influenza virus [20]. Sensitivity analyses exploring the consequences of those different assumptions were performed, with $c$ in a range 5–20 d$^{-1}$, $\mu$ in a range $10^{-5}$–$10^{-3}$ and the proportion of target cells being equal to 1% (see **S3 Text**). Thus, the estimated parameters in each compartment were $V_0$, p, β and δ.

**Statistical model.** The structural model used to describe the observed log$_{10}$ viral loads $Y_{ijk}$ of the i$^{th}$ animal at the j$^{th}$ time point in the k$^{th}$ compartment (k = 1 for nasopharyngeal or k = 2

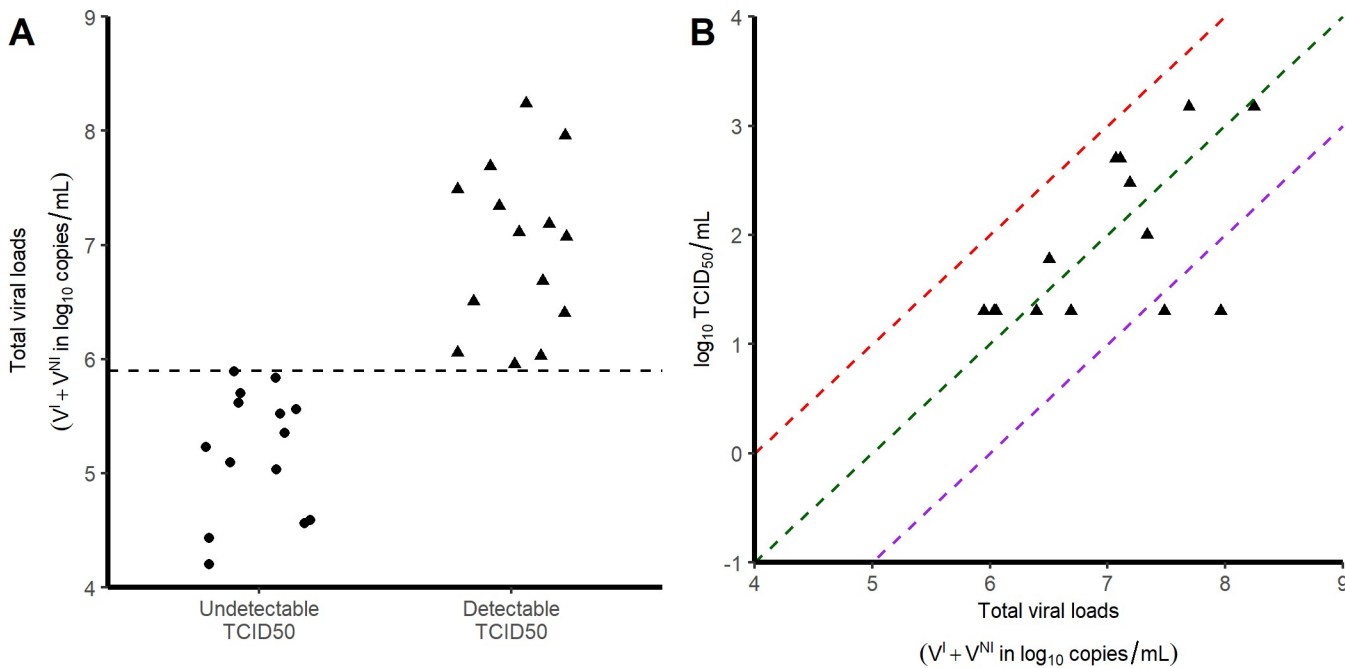

**Fig 2. Relationship between TCID$_{50}$ and viral loads. Each symbol corresponds to a tracheal swab obtained at 3 dpi.** A) Viral load levels in samples with no detectable infectious virus (circles) and those with detectable infectious virus (triangles). B) Correlation between the viral loads and the infectious virus, as measured by TCID$_{50}$. Red, green and purple dashed lines represent putative ratios of infectious virus of $10^{-4}$; $10^{-5}$ and $10^{-6}$ respectively.

for tracheal) is

$$Y_{ijk} = f(\theta_{ik}, t_{ijk}) + e_{ijk} \qquad (6)$$

Where $f$ is the function describing the total viral loads dynamics over time ($V^I(t)+V^{NI}(t)$); $\theta_{ik}$ is the vector of parameters of subject $i$ in the $k^{th}$ compartment and $e_{ijk}$ is the additive residual error. Individual parameters $\theta_{ik}$ are supposed to follow a log-normal distribution with a median value that depends on the compartment:

$$\theta_{ik} = \gamma \times exp(\eta_i) \times \exp\,(\alpha \times I_{k=2}) \qquad (7)$$

where $\gamma$ indicates the fixed effects and $\eta_i$ the individual random effects, which are supposed to follow a normal distribution of mean zero and standard deviation $\omega$, and $\exp(\alpha)$ is the vector of the ratio values between the nasal and tracheal compartments. The residual error $e_{ijk}$ is assumed to follow a normal distribution of mean zero and constant standard deviation $\sigma_k$. Standard errors were calculated by drawing parameters in the asymptotic distribution of the parameter estimators. For each parameter, we calculated the 2.5 and 97.5% percentile to derive the 95% confidence interval.

**Modelling strategy.** As a first step, we considered nasopharynx and trachea parameters as two distinct compartments and we then tested whether the virus could migrate from one compartment to the other at a constant first order migration rate g. Parameter g was set to 0 if it did not improve the fit to the data or could not be precisely estimated. In order to reduce the number of remaining parameters, we fixed the ratio of effective inoculum, $V_{0,T}/V_{0,N}$, to 9. Then we tested the possibility for estimated parameters to be equal in both nasopharynx and trachea (e.g., $\alpha = 0$) and tested then if their variances could be set to 0 (e.g., $\omega = 0$). To do this,

we used a backward selection procedure and stopped once the BIC did not decrease anymore. Lastly, based on the Empirical Bayes Estimates (EBE), we screened the random effects for correlations. Only correlations with a Pearson's coefficient >0.8 were implemented in the model. All models tested are presented and compared in **S4 Text**.

## Viral titers determination on Vero-E6 cells

Vero-E6 cells (mycoplasma-free) were seeded in 24-well plates ($2\times10^5$ cells/well) and cultured in DMEM (Thermo Fisher Scientific) containing 1% PS (Penicillin 10,000 U/mL; Streptomycin 10,000 μg/mL) supplemented with 5% FBS (Foetal Bovine Serum) and incubated at 37˚C in the presence of 5% CO2. The next day, cells were inoculated in triplicate with 100μL per well of the tracheal swab dilutions (1:2, 1:10, 1:50,1:250) in DMEM, 1% PS, 0.1% TPCK trypsin and incubated for 1 hour at 37˚C. After removal of the inoculum, 1mL of DMEM, 1% PS, 0.1% TPCK trypsin was added in each well before incubation at 37˚C in the presence of 5% $CO_2$ for 72 hrs. The presence of a cytopathic effect (CPE) was visualized under the microscope and the $TCID_{50}$ i.e. the tissue culture infective dose leading to 50% of the maximal cytopathic effect, was calculated using the method of Reed and Münch. All experiments were conducted under strict BSL3 conditions.

## Plasma cytokine analysis

In all 31 macaques, the concentration of 30 cytokines were measured at 0, 2, 5, 7 and 9 dpi. Among them, CCL11, CCL2, IFN-α, Il-15, Il-1Ra and Il-2 were of particular interest as their kinetic changed during the infection. To identify the cytokines to be incorporated in the model, we correlated the are under the cytokine curve (AUC, calculated by the linear trapezoidal method) with the AUC of $\log_{10}$ viral loads predicted by the model.

## Models assuming a compartment for the innate immune response

We considered additional models incorporating a compartment for an antigen-dependent immune response, F, given by $\frac{dF_X}{dt} = qI_{2,X} - d_F F_X$. In these models, F could either i) reduce viral infectivity ii) and iii) put target cells into a refractory state, iv) reduce viral production and v) increase the loss of infected cells (see **S5 Text**).

## Simulations of a prophylactic treatment

We used the median estimated parameter values of the model to simulate the effects of an antiviral treatment, initiated before infection, on the viral kinetics. We explored the effects of 10 to 100 fold lower viral inoculum (corresponding to doses of $10^5$ and $10^4$ pfu) as well as the effects of drugs acting on viral production, viral entry or the proportion of infectious virus. For each scenario, we also calculated the 95% confidence interval of the median time to viral clearance, using the method described above.

## Parameter estimation

Parameters were estimated with the SAEM algorithm implemented in MONOLIX software version 2018R2 allowing to handle the left censored data [23]. Likelihood was estimated using the importance sampling method and the Fisher Information Matrix (FIM) was calculated by stochastic approximation. Graphical and statistical analyses were performed using R version 3.4.3.

## Results

### SARS-CoV-2 viral kinetics

In our experiment, cynomolgus macaques developed a rapid infection, with viral loads peaking 2 days post infection (dpi) in both nasopharyngeal and tracheal compartments. Afterwards, both nasopharyngeal and tracheal viral loads rapidly declined exponentially, with a similar median rate of 1.9 d$^{-1}$, corresponding to a daily reduction of 0.8 log$_{10}$ copies/mL (**Fig 1**). Because the viral load peaked later and higher in in the nasopharynx than in the trachea (7.9 and 7.2 log$_{10}$, respectively), the median time to unquantifiable viral load nonetheless occurred later in the nasopharynx than in the trachea (9 and 7 dpi, respectively).

Overall, the area under the viral load curve (AUC) was larger in the nasopharynx than in the trachea (45 vs 38 log$_{10}$ copies.day/mL, $p<10^{-4}$). In addition to viral RNA, we also measured virus titers in Vero-E6 cells using tracheal swabs obtained at 3 dpi (see Methods). All samples contained more than 4 log$_{10}$ copies/mL however viral growth was observed only in those having more than 6 log$_{10}$ copies/mL (**Fig 2A**). In those for which viral culture could be obtained, the ratio of TCID$_{50}$ (median tissue culture infective dose) to the total number of RNA copies ranged between $10^{-4}$ to $10^{-6}$ (**Fig 2B**).

### Viral dynamic model

Importantly, there was no antiviral effect in animals receiving various doses of HCQ compared to those receiving vehicles, even after adjustment on the mean HCQ exposure [16]. Thus, the effect of treatment was neglected as a first approximation. We later challenged this hypothesis but found no significant effect of HCQ in reducing viral production or viral infectivity (see **S1 Text**).

Physiologically, viruses can migrate from the nasopharynx to the trachea through respiration and movements of ciliary cells at the mucosal surface. Thus, we tested the possibility for viruses to move from nasopharyngeal to tracheal compartment and *vice versa* by linking both with a bidirectional rate constant $g$. However, possibly due to data paucity, the parameter $g$ was not significantly different from 0 (CI$_{95\%}$ included 0) and was therefore set to 0 in the following. Then, using a backward selection procedure we found that the infectivity rate $\beta$ and the viral production $p$ were different between nasopharyngeal and tracheal compartments (**Table 1**). The final model well fitted the data and allowed the estimation of several parameters of the infection (**Fig 3**). We estimated the effective initial viral load to $1.7\times10^{6}$ and $8.0\times10^{7}$ copies/mL in tracheal and nasal compartments, respectively, which corresponds to a total inoculum of approximately $10^{8}$ total RNA, i.e., about 1% of the total injected dose (see Methods). We found estimates of $\beta$ of $1.2\times10^{-3}$ (CI$_{95\%}$ $0.5\times10^{-3}$–$2.8\times10^{-3}$) and $1.8\times10^{-3}$ (CI$_{95\%}$ $0.6\times10^{-3}$–$5.6\times10^{-3}$) mL/virion/day ($p<10^{-4}$) in nasopharynx and trachea, respectively, while $p$ was estimated to $4.8\times10^{4}$ (CI$_{95\%}$ $1$–$8.8\times10^{4}$) and $2.2\times10^{4}$ (CI$_{95\%}$ $0.7$–$4.1\times10^{4}$) virions/cell/day ($p<0.05$). Consequently the product $p\times T_0$ was equal to $9.5\times10^{8}$ (CI$_{95\%}$ $2.0$–$17\times10^{8}$) and $3.9\times10^{8}$ (CI$_{95\%}$ $1.2$–$17.7\times10^{8}$) virions/mL/day in nasopharynx and trachea, respectively.

The loss rate of infected cells, $\delta$, was not found different between the two compartments and estimated to 1.9 (CI$_{95\%}$ 1.6–2.3) d$^{-1}$ corresponding to an infected cell half-life of 9 (CI$_{95\%}$ 7–13) hours. These parameter estimates allow us to derive the basic reproduction number $R_0$ corresponding to the number of infected cells generated by a single infected cell at the beginning of the infection. We found $R_{0,N}$ equal to 5.6 (CI$_{95\%}$ 1.3–21) and $R_{0,T}$ equal to 3.8 (CI$_{95\%}$ 0.7–18) in the nasopharynx and the trachea, respectively. One can also derive the viral burst size $N$ corresponding to the number of viruses produced by an infected cell over its lifespan. We found $N_N$ = 25,000 (CI$_{95\%}$ 5,900–72,000) virions and $N_T$ = 11,000 (CI$_{95\%}$ 4,000–19,000) virions.

**Table 1. Population parameter estimates of the final model described by Eq (1) to (5).** Numbers in parenthesis are the relative standard error expressed in percentage (RSE%) associated either to the fixed or the standard deviation (SD) of random effects.

| Parameters (units) | Fixed effects (RSE%) | SD of random effects (RSE%) |
|---|---|---|
| $\beta_T$ (mL.copie/d) | $1.8\times10^{-3}$ (42) | 0.3 (33) |
| $\beta_N$ (mL.copie/d) | $1.2\times10^{-3}$ (12) | |
| $p_T$ (copies/cell/d) | $2.2\times10^{4}$ (49) | 1.0 (27) |
| $p_N$ (copies/cell/d) | $4.8\times10^{4}$ (40) | |
| $V_{T,0}$ (copies/mL) | $8.0\times10^{7}$ (9) | - |
| $V_{N,0}$ (copies/mL) | $1.7\times10^{6}$ (9) | - |
| $\delta$ (1/d) | 1.9 (9) | 0.2 (37) |
| c (1/d) | 10 (fixed) | - |
| $\mu$ (unitless) | $10^{-3}$ (fixed) | - |
| k (1/d) | 3 (fixed) | - |
| $T_{T,0}$ (cells/mL) | $1.88\times10^{4}$ (fixed) | - |
| $T_{N,0}$ (cells/mL) | $1.98\times10^{4}$ (fixed) | |
| $\sigma_T$ | 1.06 (6) | - |
| $\sigma_N$ | 1.19 (6) | - |

## Sensitivity analysis

In our main analysis, the viral clearance c and the proportion of infectious virus $\mu$ were fixed. We tested the robustness of our results to different values of these parameters (see **S3 Text**) and we used model averaging (MA) to compute the averaged parameters values following a methodology presented in [24]. Overall, models were broadly undistinguishable as they provided a Bayesian Information Criterion within a 4 points range. Model averaged parameter estimates of $\delta$, $R_{0,N}$ and $R_{0,T}$ were equal to 1.9 ($CI_{95\%}$ 1.5–2.3) d$^{-1}$, 6.8 ($CI_{95\%}$ 1.3–29) and 4.4 ($CI_{95\%}$ 1–26), respectively.

## Immune markers during SARS-CoV-2 infection

Among the 30 cytokines tested, 6 greatly varied during the infection and peaked at 2 dpi (namely CCL11, CCL2, IFN-$\alpha$, IL-15, IL-1RA and IL-2) but there was no association between cytokine and viral loads (**Fig 4**). A model assuming an effect of the immune response in cell protection resulted in a reduced BIC of 6 points. However, the gain in fitting criterion was entirely due to 3 individuals (see **S5 Text**) and led to more uncertainty in parameter estimates due to increased complexity (see **Table B in S5 Text**). Moreover none of the cytokines measured during the experiments, including IFN-a, showed a correlation with viral dynamics (**Fig 5**). Thus, overall, there was no clear evidence in these data that a more complex model improved the understanding of viral dynamics over a simple target cell model.

## Expected profiles with prophylaxis treatments

We used estimated parameter values of the model to simulate the effects of an antiviral treatment, initiated before infection, on the viral kinetics. We explored different viral inoculum (ranging from $10^4$ to $10^6$ pfu), drugs mechanisms of action (blocking viral production, viral entry, or infectious virus production), with different levels of antiviral efficacy. As the conclusions were not sensitive to the mechanism of action, we present below the results for a prophylactic drug blocking viral entry (**Fig 6**). Other mechanisms of action are presented in **S6 Text.**

In both compartments, the median time to viral clearance increased with lower doses of inoculum, and ranged from 10.1 ($CI_{95\%}$ 8.4–14.7) days with $10^6$ pfu to 11.7 ($CI_{95\%}$ 9.5–19.0)

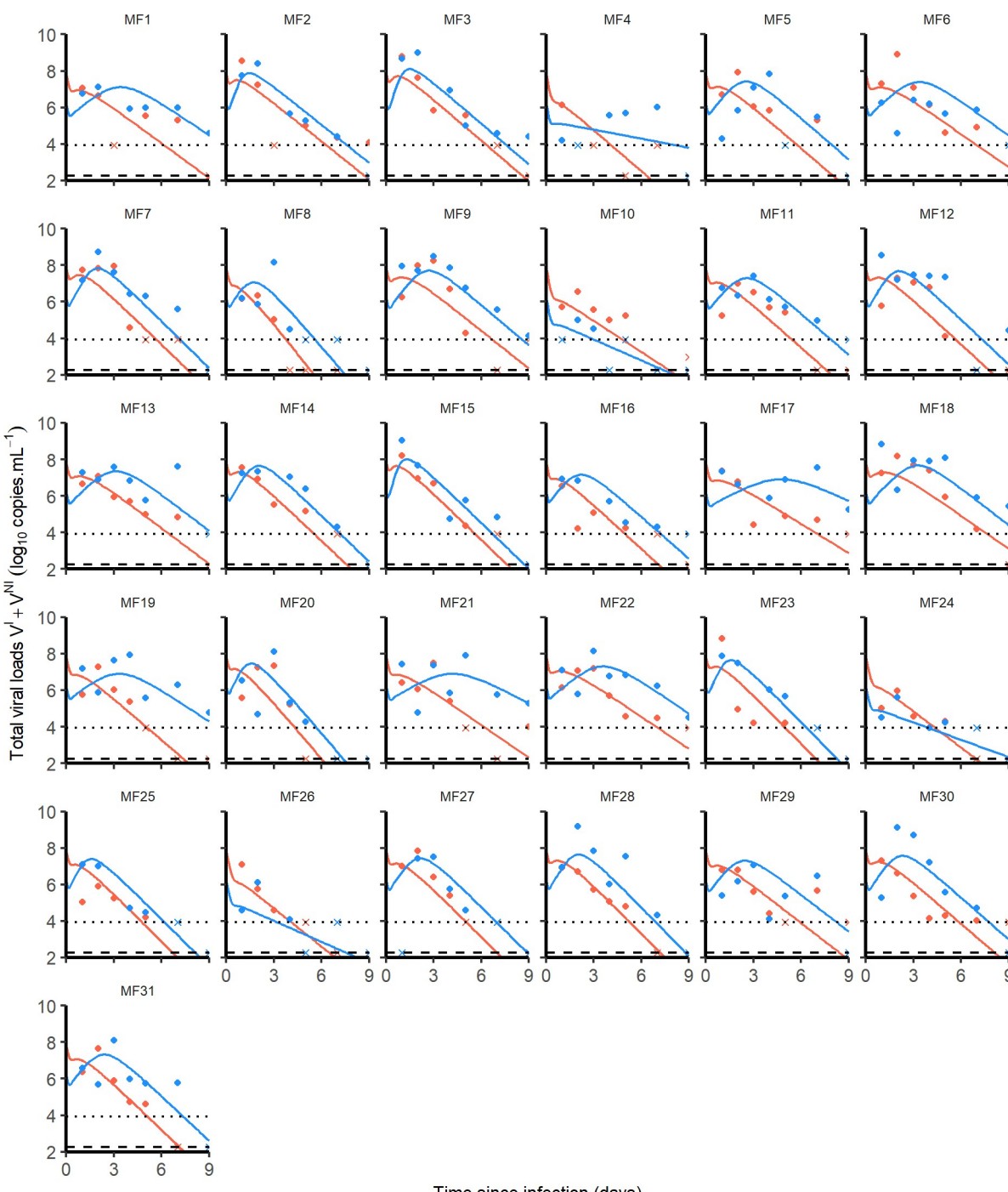

**Fig 3.** Nasopharyngeal (blue) and tracheal (red) individual predicted dynamics by the model described in Eqs (1) to (5). Full dots are the quantifiable viral loads and crosses the observation below the limit of quantification. The dotted line represents the limit of quantification and the dashed line the limit of detection.

days with $10^4$ pfu in the nasopharynx. A 90% effective antiviral treatment administered upon infection would dramatically reduce peak viral load in all scenario and maintain virus below the threshold level of infectivity of 6 $\log_{10}$ copies/mL (**Fig 2A**). A 99% effective antiviral treatment could in addition abrogate viral load viral load, with time to viral clearance ranging from

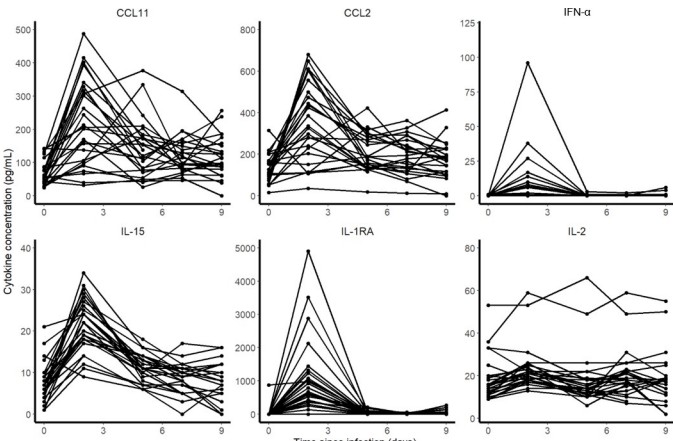

**Fig 4. Cytokine concentrations during SARS-CoV-2 infection.**

3.5 ($CI_{95\%}$ 1.9–6.2) days with $10^6$ pfu to 0.5 ($CI_{95\%}$ 0.2–2.5) days with $10^4$ pfu in the nasopharynx.

## Discussion

We here developed a mathematical model for SARS-CoV-2 viral dynamics using nasopharyngeal and tracheal swabs obtained in 31 infected macaques. Using this model we could estimate key parameters of virus pathogenesis, in particular the viral infectivity rate (equal to $1.2 \times 10^{-3}$ and $1.8 \times 10^{-3}$ in tracheal and nasopharyngeal compartments, respectively), and the loss rate of infected cells, estimated to 1.9 $d^{-1}$ in both compartments (e.g., a half-life of 9 hours). Consequently, we estimated the number of secondary cell infection resulting from one infected cells, $R_0$, to approximately 4 and 6 in tracheal and nasopharyngeal compartments, respectively. This value of $R_0$, together with the large viral inoculum used in this experimental model ($10^6$ pfu), explains that tracheal viral loads barely increased post-infection and that nasopharyngeal viral loads rapidly peaked at 3 dpi. After peak viral load, the rapid loss rate of infected cells was sufficient to explain that clearance of the virus occurred around day 10 in both compartments, and we did not find evidence in this model for a role of an immune response mediated by cytokines in accelerating the viral clearance.

Although the number of animals and the very detailed kinetic data allowed the precise estimation of several parameters, some limitations exist. First, animals were infected with a large inoculum ($10^6$ pfu, i.e. $10^{10}$ RNA copies) which rapidly saturate target cells making the estimation of β less robust and difficult to distinguish the processes of clearance of the inoculum from those of *de novo* viral infections. In the future, the analysis of subgenomic RNA, that quantifies intracellular viral transcription, will provide important information to distinguish these two processes and provide a more precise estimate of $R_0$ [25,26]. Second, a number of unknown parameters were fixed to ensure identifiability, in particular the proportion of infectious virus, $\mu$, and the number of target cells available in each compartment, $T_0$. In this model, $p \times T_0$ is the only reliable quantity since the estimate of p depends on the number of susceptible cells. We here estimated the number of alveolar type II cells by analysis of the size of the tissues of one euthanized animal and we assumed that 0.1% of these cells could be target for the virus. Although this value led to more coherent parameter estimates (see **S3 Text**), this is higher than the value of 1% of cells expressing the ACE2 receptor and serine protease TMPRSS2 found in humans [27,28]. Specific experiments will be needed to estimate the proportion of target cells

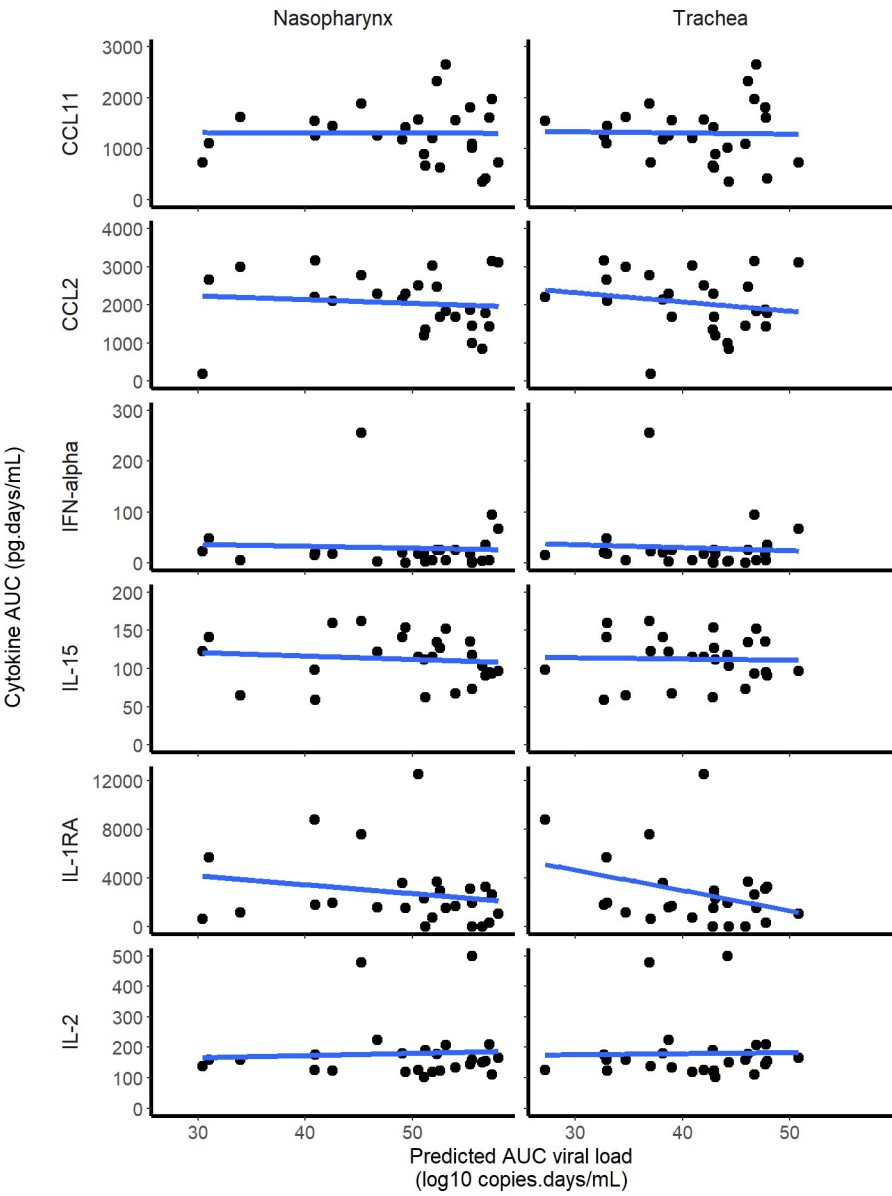

**Fig 5. Correlation between the individual predicted AUC log$_{10}$ viral load and cytokine AUC.** None of the 6 cytokines that increased during the infection was significantly correlated with the predicted viral load AUC in the nasopharynx or the trachea.

in the URT of macaques. Third, we relied only on measures in both compartments of the upper respiratory tract, which may not reflect the kinetics in the lower respiratory tract. It is in particular possible that the kinetics of both the virus and the immune response may be different in the lung, and that both cytokine responses and the lesions as observed by CT scans may be associated with viral loads in the lungs. In our experiments the first viral load measurements in bronchoalveolar lavages (BAL) was made at 6 dpi, and were all below the limit of detection at the next available data point at day 14, precluding a more detailed analysis of the kinetics in the lungs.

We also aimed to evaluate whether more complex models including an antigen-dependent immune response could improve the data fitting. A model assuming that cells could be put in

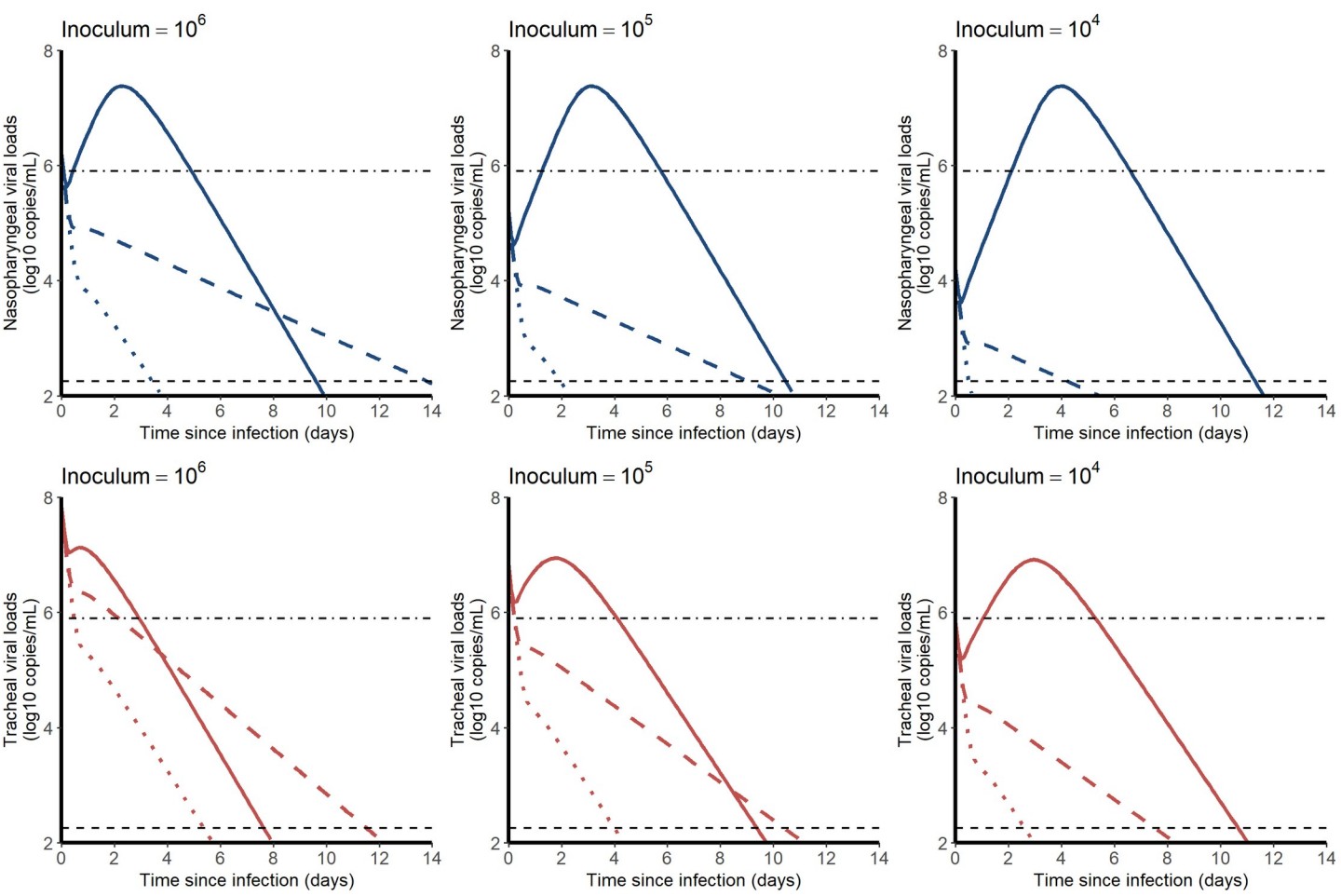

**Fig 6. Median viral kinetic profiles in the nasopharynx (blue) and the trachea (red) according to the inoculum size and the level of an antiviral initiated in prophylaxis and blocking viral entry β.** Treatment efficacy of 0% (no treatment, solid line), 90% (dashed line) and 99% (dotted lines) were considered. Point-dotted line represents the threshold below which no virus could be cultured in vitro (**Fig 2**).

an antiviral state improved the BIC (see **S4 Text**), however this improvement was due to 3 individuals, and this came to the expense of a deteriorated precision of parameter estimates. Further, none of the observed cytokines were found associated with viral dynamics (**Fig 5**), suggesting that this improvement in data fitting was not supported by our data. In addition, none of the animals had detectable antibodies until day 7, and only 25% had detectable antibodies by day 14, suggesting that the humoral response played a minor role in viral clearance. The role of the immune response in this experimental model of mild infection is unclear, but our findings are consistent with data obtained in patients with an asymptomatic infection, in which the immune response and the cytokine response remained low throughout the infection period [29]. Accordingly our estimate of the duration of viral shedding was between 10 and 12 days in the nasopharynx, depending on the initial inoculum, very close to the values of 9 days (*Diamond Princess* [30]) and 12 days [11] estimated in mild or asymptomatic individuals. From an experimental setting, the analysis of viral dynamics in animals infected with different size of viral inoculum could also bring insights on the need to use more complex models [31].

Finally, we used the model to inform on the use of prophylactic drugs in this macaque model. Given our estimate of $R_0$, a 90% effective treatment should be able to prevent virus

growth and would maintain the viral load levels below $10^6$ copies/mL at all times, making the chance of detecting infectious virus very limited, consistent with our modeling predictions in humans ([32]).

## Supporting information

**S1 Text. Effects of hydroxychloroquine.**
(DOCX)

**S2 Text. Model file for Monolix.**
(TXT)

**S3 Text. Sensitivity analysis.**
(DOCX)

**S4 Text. Model building.**
(DOCX)

**S5 Text. Immune response models.**
(DOCX)

**S6 Text. Simulations.**
(DOCX)

**S1 Data. Nasaopharyngeal and tracheal swabs data file.**
(TXT)

## Acknowledgments

We thank Peter Czuppon (Collège de France & Sorbonne Université), François Blanquart (Collège de France & INSERM UMR1137) for critical reading and expertise on the analysis.

## Author Contributions

**Conceptualization:** Antonio Gonçalves, Pauline Maisonnasse, Vanessa Contreras, Thibaut Naninck, Romain Marlin, Caroline Solas, Xavier de Lamballerie, France Mentré, Roger Le Grand, Sylvie van der Werf, Jérémie Guedj.

**Data curation:** Antonio Gonçalves, Pauline Maisonnasse, Flora Donati, Mélanie Albert, Sylvie Behillil, Vanessa Contreras, Thibaut Naninck, Romain Marlin, Caroline Solas, Andres Pizzorno, Julien Lemaitre, Nidhal Kahlaoui, Olivier Terrier, Raphael Ho Tsong Fang, Vincent Enouf, Nathalie Dereuddre-Bosquet, Angela Brisebarre, Franck Touret, Catherine Chapon, Bruno Hoen, Bruno Lina, Manuel Rosa Calatrava, Xavier de Lamballerie, France Mentré, Roger Le Grand, Sylvie van der Werf.

**Formal analysis:** Antonio Gonçalves, Pauline Maisonnasse, Vanessa Contreras, Thibaut Naninck, Romain Marlin, Caroline Solas, Xavier de Lamballerie, Roger Le Grand, Sylvie van der Werf, Jérémie Guedj.

**Funding acquisition:** Jérémie Guedj.

**Investigation:** Antonio Gonçalves, Pauline Maisonnasse, Vanessa Contreras, Romain Marlin, Caroline Solas, Julien Lemaitre, Nidhal Kahlaoui, Xavier de Lamballerie, France Mentré, Roger Le Grand, Sylvie van der Werf, Jérémie Guedj.

**Methodology:** Antonio Gonçalves, Pauline Maisonnasse, Vanessa Contreras, Thibaut Naninck, Romain Marlin, Caroline Solas, Xavier de Lamballerie, France Mentré, Roger Le Grand, Sylvie van der Werf, Jérémie Guedj.

**Project administration:** Pauline Maisonnasse, Roger Le Grand, Sylvie van der Werf, Jérémie Guedj.

**Supervision:** Raphael Ho Tsong Fang, Nathalie Dereuddre-Bosquet, Catherine Chapon, France Mentré, Sylvie van der Werf, Jérémie Guedj.

**Writing – original draft:** Antonio Gonçalves, Pauline Maisonnasse, Vanessa Contreras, Thibaut Naninck, Caroline Solas, Roger Le Grand, Sylvie van der Werf, Jérémie Guedj.

**Writing – review & editing:** Antonio Gonçalves, Pauline Maisonnasse, Vanessa Contreras, Thibaut Naninck, Caroline Solas, Roger Le Grand, Sylvie van der Werf, Jérémie Guedj.

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
