## [Decision Letter · Decision Letter 0]

22 Nov 2020

Dear Mr. Gonçalves,

Thank you very much for submitting your manuscript "SARS-CoV-2 viral dynamics in non-human primates" for consideration at PLOS Computational Biology. As with all papers reviewed by the journal, your manuscript was reviewed by members of the editorial board and by several independent reviewers. The reviewers appreciated the attention to an important topic. Based on the reviews, we are likely to accept this manuscript for publication, providing that you modify the manuscript according to the review recommendations.

The reviewers agreed that your work would be a great contribution to the current literature on SARS-CoV-2 and COVID-19. However, the reviewers raised many, in our opinion valid and valuable, points of criticism. We encourage you to address these in your revision as we believe that this strengthen your paper and increase its impact. Especially the point raised by Reviewer #3 on the very high challenge dose in the experiments and the complications that introduces for the estimation of R0 would be important to address. Reviewer #1 asks you to explore additional mechanisms that might be able to give you a better model fits (point 1). Reviewer #4 ask you to further justify some model assumptions (point 5). In our view, the paper would gain if you justified which mechanisms you chose to include and which potential mechanisms you omitted.

Sincerely,

Roland R Regoes

Associate Editor

PLOS Computational Biology

Rob De Boer

Deputy Editor

PLOS Computational Biology

[LINK]

Reviewer's Responses to Questions

**Comments to the Authors:**

Reviewer #1: To characterize the SARS-CoV-2 nasopharyngeal and tracheal viral load dynamics in 31 cynomolgus macaques, the authors developed a target-cell limited mathematical model for each tissue compartment. The model includes an eclipse phase and the possibility of production of non-infectious viruses. The model is used to estimate basic reproductive ratios for each tissue compartment and other kinetics parameters. The authors also performed correlation tests between immune markers with viral load and explored competing models that include association of viral load and IFN-α levels. They finally made projections of the effect of >90% effective prophylactic interventions. The manuscript is well written and the model is based on a well-known model of virus dynamics. My biggest concern is related to the accuracy of the model fits in capturing the observed behavior, and therefore, the accuracy of parameter estimates and conclusions based on the model.

Major comments

1. It looks like the viral load observations have different kinetic patterns beyond a viral expansion and clearance, and therefore, the model is not able to capture them. This might be either due to high measurement error or to an additional mechanism that is not in the model (or its extensions using the IFN-α-related models). For example, for the nasopharyngeal observations (blue in Fig 4) there is the possibility of more than one episode/peak (MF6, 13,17,18,19, 21,29) or a plateau/sustained phase where the virus is maintained for several days (MF1, 6, 12 and some of the others mentioned before). In that sense, it is not clear if this phrase in page 5 is completely accurate: “After the peak, both nasopharyngeal and tracheal viral loads rapidly declined exponentially “. It also looks like the model doesn’t capture the nasal dynamics of animals MF4, 5, 10, 26. I suspect that adaptations of this model structure that allows dynamics with a sustained viral load, that may add a couple of parameters could give a significantly lower BIC. I think a larger exploration of models has to be performed to confirm the parameter estimation and conclusions based on the model fits.

2. The paper discusses that the viral peak happens about 2 days post-infection. But there is not a discussion of why the model cannot capture the viral peak and that 2-day time.

3. There is not a discussion of the reasons to select the values of V0 presented in Table 1. There is only a citation to reference 20 (Long Q-X et al, Nat Med) without giving any reason. I wonder how the model would behave using different values of V0. It is possible that allowing for a lower V0 (or other model structures) can capture the dynamics of some of the nasal viral loads that had low levels or its behavior seems delayed (MF4, 5, 10, 24, 26).

Minor comments

4. It was not completely clear if the %RSE in Table 1 was related to the fixed or random effects of the statistical model.

5. There is a missing “)” at the end of page 6 when describing the estimates of βn.

6. The words “we tested” are repeated in the 4th line in the modeling strategy section in page 17.

Reviewer #2: Goncalves et al. previously reported on their results of a study in non human primates and created a dataset of longitudinal virus load data in the upper respiratory tract (URT) for 31 cynomolgus macaques in response to mild SARS-CoV-2 infections. Here, the authors perform mathematical modeling and, using their data, estimate viral kinetics parameters and hypothesize that potent prophylactic treatment can drive the virus to extinction without prior viral growth. The relatively simple model used in their manuscript was compared against more complicated models including virus migration between nasopharyngeal and tracheal compartment, and cytokine mediated immune responses. These models were not found to explain the data significantly better hinting at a minor role of migration and immune response in the URT during mild SARS-CoV-2 infections.

Longitudinal virus load data, in particular data including the pre-symptomatic phase, are extremely rare and hence, despite being limited to the URT and mild infections, the authors study provides an important dataset and estimates for viral kinetics parameters.

However, there are several issues that require attention.

1) You report a virus threshold below which viral culture could not be obtained which is a very interesting result. In Fig. 2B, you compare the ratio of TCID50 and viral load with the viral load (“In those for which viral culture could [be] obtained, the ratio of TCID50 to the number of RNA copies ranged between 10-4 to 10-6 (Fig. 2B). “). It is not clear to me what the importance/interpretation of this graph is. Can you please elaborate on this. Are only the ratios important or is the comparison to the viral load important (e.g. correlated/uncorrelated)? Why does this ratio represent an underestimate of the total infectious virus? (p.16 (“the ratio of titers to RNA copies represent an underestimate of the total infectious virus ”))

2) In the model you split the virus into infectious and uninfectious virus. For clarification it may be beneficial to label viral load as total viral load (V^I + V^NI) in your figures. Furthermore, there are some other issues related to the viral load(s) that require attention (Table 1).

- Why is the unit of initial conditions V_T0 and V_N0 copies and not cp/ml? As is, with the given units for beta and T_T(t=0), T_N(t=0), the units in eq. (1) don’t work out.

- If V_T0 and V_N0 are initial conditions for infectious virus only, then label as V^I_T0, V^I_N0. Furthermore, I do not understand how you obtain the values for V_T0 and V_N0, given that (i) the viral inoculation is 10^10 copies (discussion), (ii) 90% (4.5/5ml) of the inoculum are applied to the tracheal and 10% (0.5/5ml) to the nasopharyngeal compartment and (iii) a fraction mu = 10^(-3) of virions is assumed to be infectious.

Other minor issues

3) In the experiments, is there a reason for splitting the virus inoculation 90% tracheal vs 10% nasopharyngeal? Is this supported by literature of how virus enters the URT?

4) (p.5) “After the peak, both nasopharyngeal and tracheal viral loads rapidly declined exponentially, with a median rate of 0.8 log10 copies/mL every day (Fig. 1). The slope in viral decline was more rapid in the trachea than in the nasopharynx, leading to a first measurement below the limit of quantification (LOQ=8514 copies/mL) 7 and 9 dpi, respectively. “

- Is the 0.8 log10 copies/ml the combined median value for nasopharyngeal and tracheal viral loads?

- Given figure 1 it seems to me that the first measurements below LOQ are much earlier than the reported 7 and 9 dpi. Are these average/median values? Please clarify.

5) (p.6) Equation (2): Typo (VI_X instead of V^I_X)

6) (p.7) “Assuming that only 0.1% of cells express ACE2 receptors at their surface (17) “

- According to figure 1b) in ref. (17) approximately 1% of epithelial cells express ACE2. To my understanding figure 1a) from where I assume you extracted the 0.1% shows all cell types. But since you only model epithelial cells I believe you should work with 1%. Since this will change your T_0 estimates, it will probably also have an impact on p and p*T_0.

7) ”We found R0,N equal to 5.9 (2.0 – 16) and R0,T equal to 4.0 (1 – 13). Together with the high inoculum in the trachea (see methods), the viral load scarcely increased in the trachea while clearly increased until 3 dpi in the nasopharynx (Fig. 4). One can also derive the viral burst size N corresponding to the number of viruses produced by an infected cell over its lifespan. We found NN = 22,000 (8,100 – 36,000) virions and NT = 10,000 (4,500 – 19,000) virions.”

- Please refer to the methods section where you give the formulas for R_0 and N.

8) (p.10, Discussion) “we could estimate key parameters of virus pathogenesis, in particular the production rate from infected cells (equal to 1.9 × 104 vs 3.6 × 104 virions/cell/day in tracheal and nasopharyngeal compartments, respectively)”

- You say twice (in the results and the methods) that only p*T0 can be identified. For clarification you should state in the discussion that these estimates are obtained for fixed T0. This seems particularly important to me since you claim a certain degree of uncertainty in your T0 estimate on p.11.

9) (p.16, Statistical model) How is the function f defined? (How) does it relate to model equations (1)-(5)? Please explain.

10) (Supplementary Information file 2) “As the viral clearance c and the eclipse phase rate µ could not be estimated from the data”.

- Mu is not the eclipse phase but the fraction of infectious virus.

11) The manuscript should be checked for typos.

Reviewer #3: In this study Goncalves et al analyze the kinetics of viral load during the entire course of SARS-CoV-2 infection in non-human primates. They describe the dynamics with a mathematical model and estimate model parameters by fitting to data. They also collect longitudinal levels of a panel of cytokines involved in the innate immune response and test for association between these markers and viral load kinetics.

There are many very strong and unique aspects of this study, which I think will make it quite valuable for the field. Longitudinal within-host viral load kinetics are still very poorly characterized for COVID-19, since it is very difficult to catch human subjects early in infection (before symptom onset) and sample them frequently. For this reason, the full timecourse of viral load, its variation between individuals, and its association with the disease course are still not clear. Thus, animal models are still very useful. This study represents a very large cohort of animals with very frequent sampling and a large number of biomarkers measured, so it helps fill in some of these gaps. The authors use a mechanistic mathematical model to describe the data, as opposed to just fitting a simple curve, which has the benefit that it then be used to examine how kinetics might change under different parameter perturbations. They use a rigorous statistical method for group-level fitting, which make it easy to then use the model to simulate expected inter-patient variance in kinetics. The availability of so many immunological markers measured alongside viral loads is unique, and the authors use some very nice methods to test for the impact of these markers on viral kinetics. The Discussion section does a very thorough job of describing the many caveats of their work.

I think there are two major limitations to this work. I do not think the authors can or should do anything differently to address them, I am simply pointing them out to help with the editorial decision.

The first is about the study design in the animal model. It is not clear how much this is helping us understand human infection, because the animals seem to be inoculated with an extraordinarily high dose of virus, and in nearly all animals the entire upslope of the viral load is missing (just like it generally is in human data collected post-symptom onset). We know for many viral infections inoculum size influences disease course, so its possible this high-dose infection model might be qualitatively different than a more realistic infection scenario. And, with the upslope missing, it is nearly impossible to accurately estimate R0 - for infection models within-host or at the population level, it is the early exponential growth phase that provides the information that allows R0 to be an identifiable parameter combination (assuming you separately know/can estimate the lifespan of infected cells). Also, I worry that this initial “overdose” of virus is obscuring some inter-patient variation that exists in natural infection. In humans, there seems to be considerable variation in the time to symptom on set (and therefore likely also in viral load peak), whereas in this model viral load peaks extremely quickly and at a similar time in all animals, so it is unclear how relevant this model is.

The second limitation is that there is really not much about this data that motivates the need to apply a mathematical model, and even more specifically the quite complicated model the authors used. I myself am a modeler and do not need to be convinced of the value of models in general … it’s just that here it sort of seems like a model was used just for the sake of using a model. In the paper, Figure 1 is shown, and then the authors launch into a complex model description. But why? When I look at Figure 1, nothing suggests to me you would need or want to apply any sort of mathematical model to this data. It just looks like decay over time. If you are going to write a whole paper about applying a viral dynamics model to data, it would be helpful to provide the reader with some motivation for that! And the model seems relatively complex - again, you need to motivate this. What is the biology you think is going on and why did you decide to capture it this way? How are the nasopharyngeal and tracheal compartments related, physiologically? How would you possibly have enough information to identify migration of cells? Why did you feel the need to include infectious and non-infectious virus? What is the biology behind this and why do you think it’s necessary to explain your data? Why do you think this process is target cell limited? Where did these #s of cells come from? Why are non-productive cells needed? Then looking at Figure 4, honestly the model does not seem to be doing a particularly amazing job explaining any of this data. Mostly, the data seems to be all over the place and barely following a trend. The model seems to be missing the peak in most animals, and not accurately capturing the early kinetics in the few animals for which this info is available.

Despite these concerns, I do still feel like the results of the paper are useful, because we are still very early on in our understanding of this virus and understandably animal models are not yet perfected, and the model the authors use is at least reasonable so can provide a starting point for other work that probes these mechanisms more carefully, and for now can at minimum serve as a calibrated simulation tool for recreating within-host dynamics.

A few more specific minor comments, mainly about typos/grammar:

* Author summary: Typo in these sentences

* "We found that viral load rapidly declined after peak viral load [despite we found] no association between model parameters and immune markers.”

* The within-host reproductive basic reproduction number was estimated to BE 6 and 4 in nasopharynx and trachea suggesting that a prophylactic therapy blocking viral entry or production with 90 efficacy could be sufficient to prevent viral growth and peak viral load.

* Introduction:

* "This is due to the fact that many studies rely on large transversal analyses with few patients having serial data points or, in contrary, on detailed small series of patients,..”. I think instead of “transversal” you mean “cross-sectional”

* Methods:

* The paper really does not make sense as it is written, because you jump into Results before explaining the methods. This doesn’t work for a modeling paper. Please include all the model motivation and details that are now at the very end of the paper BEFORE the results section. It is impossible to interpret the results without understanding why you chose the model you did, what all the parameters mean, if you did group level fitting or individual level, which parameters were fixed vs fit, and why were some values fixed and what source did you use to estimate them?

* I did not understand why it was necessary to include non-infectious virus in the model (I don’t think you are fitting to data that breaks down virus this way), and I did not understand the sentence “Since the ratio of titers to RNA copies represent an underestimate of the total infectious virus and that this ratio ranged from 10-6 to 10-4 (Fig. 2), we fixed μ to 10-3 "

* Results:

* "In those for which viral culture could obtained, the ratio of TCID50 to the number of RNA copies ranged between 10-4 to 10-6 (Fig. 2B).” What is TCID50 ? This term was never defined

* Figure 1: This figure is pretty useless - it just looks like a pile of points all on top of each other and you can’t follow one animal over time. I think the authors should be able to come up with a better way to show this data. For example, different color lines/points for each animal. And maybe also showing the group mean/median over time.

* Figure 2B: don’t understand what is being shown here. It seems like the same measure (viral load) is being used in both the x and y axis. And again, what is TCID50?

* Figure 3: This figure caption is short and non-informative. It should describe the model and what each of the parameters mean.

* Figure 6: It is obvious from looking at these plots that there is there is no correlation here. I am not sure it is necessary to report all these statistics

* Table 1: What does RSE% mean? Abbreviations should be explained. Table should have another column stating the description of each parameter. The meanings of most of these parameters aren’t explained anywhere - not in the text, or figure caption.

* Re inoculum size - why is it predicted to take longer for viral clearance with lower innoculum? Would be nice to offer an intuitive explanation, since I found it surprising! Is this because of your assumption that target cell limitation is only mechanism of control? If so, it might be a good experimental test of target cell limited model - try smaller innoculums experimentally and see what happens!

* Discussion: Sentences with grammar problems

* “This hypothesis is supported by the fact that in humans 0.1% of alveolar type II cells expressing the ACE2 receptor, gate for SARS-CoV-2 to enter host cells”

* "Such estimate is unknown to our knowledge in cynomolgus macaques”

* And I don’t understand what this means: "…we nonetheless note that it leads to coherent parameter estimates verifying the condition of >R0 (i.e., the burst size of the infectious virus is larger than the number of secondary cell infection) "

Final note: When submitting a paper for review, the authors should include figures in the main text near where they are first cited, with the figure captions directly below the images. It is very inconsiderate to reviewers to expect them to jump back and forth between different sections of the paper to try to interpret figures. Also, some of the images are bad quality, which makes it very difficult to interpret the results

Reviewer #4: The authors use a simple within-host model to describe the kinetics of SARS-CoV-2. Although the model itself is not new, the authors make use of both in vivo and in vitro data to estimate important parameters (e.g. R0 and the lifespan of infected cells) and distinguish likely infectious virus from non-infectious virus. Overall the manuscript is clear and well-written though lacking in detail in a few places.

Major points:

1. The authors assume HCQ reduces viral production, but are there other possible mechanisms (e.g. reducing the transmission rate, Supplementary File 1, Fig S1), and if so, does accounting for these support the assumption of no effect of HCQ on viral dynamics?

2. Results page 10: The authors focus on drugs with 90 and 99% efficacy and identify 90% as a target efficacy for future drug development that would prevent infection. But were lower efficacies investigated? For example, was 90% the lowest efficacy at which infection was prevented? Even if so, drugs with lower efficacies may still limit infection to a significant enough degree that it would be beneficial to include them as potential targets.

3. Discussion: It would informative to discuss how the parameter estimates compare to those of other within-host models for SARS-CoV-2 (at least in human studies, there have been a wide range of reported R0 and infected cell lifespan estimates).

4. Methods: To improve reproducibility, more information on the fitting procedure is needed, for example, on the backward selection procedure, and the implementation in Monolix (e.g. initial estimates). These could be included as supplementary material. I would encourage the authors to share their code for complete reproducibility.

5. Given that infection is mild here it seems perhaps unlikely that infection is limited by depletion of all susceptible cells which - despite their assumed low abundance — would likely still involve considerable damage and inflammation. It seems at least equally plausible that the infection is limited by innate immune mechanisms or NAbs. Could you justify the choice of a target cell limited model a priori? Does the choice of model influence the key conclusions? Some discussion of this issue would strengthen the paper.

Minor points:

1. p7 - p x T0 is referred to as total viral production — this seems a little misleading as it’s never the case that all T0 targets are simultaneously infected. p15 - you say “First, the term p×T0 is the only identifiable quantity in our model, “… this is a typo - I presume you’re saying that only this combination is identifiable, not p and T0 individually.

2. Results of prophylaxis treatment, page 10: The authors investigate different levels of viral inoculum, but it is not clear how these levels, quoted in PFU, were translated to V(0), quoted in copies (e.g. Table 1), to perform the simulations?

3. Model comparisons: In model comparison tables (e.g. Supplementary FiIe 3, Table S1) it would be easier for the viewer to distinguish between model support by quoting the difference in BIC between each model and the reference model (rather than the absolute BIC value, which is meaningless)

4. Fig 4: Some macaques, nasal virus load persists for some time and there is even a suggestion of a second peak (e.g. MF13, MF17, MF29). Can the authors speculate as to why this may be (in some cases these later measurements appear to be above the threshold for obtaining infectious virus)?

5. Fig 7: It would be useful to the reader to add a horizontal line to each panel marking the threshold below which infectious virus could not be obtained in the in vitro experiments.

6. Fig 7 bottom row panels (tracheal viral loads): As I understand things, the solid curves represent dynamics for a drug with 0% efficacy (i.e. no drug) and should be the same regardless of mode of action. In other words, for each viral inoculum panel in Fig 7, the solid curve should be the same as that in the corresponding viral inoculum panel in Figures S1 and S2 (Supplementary File 4). However, this does not seem to be the case (e.g. the solid curves in Fig 7 appear to peak later than those in Figs S1 and S2)?

More generally, the authors should show somewhere in the main text or supplementary material how each mode of action is incorporated into the model equations.

7. General comment: Did the authors find any differences in dynamics between male and female macaques? As differences have been reported with respect to human infection, it would be interesting to know if any differences are apparent in the macaque data.

**Have all data underlying the figures and results presented in the manuscript been provided?**

Reviewer #1: Yes

Reviewer #2: None

Reviewer #3: **No: **The authors just say "data available upon request" but they should provide the data in spreadsheets with the SI or on a public repository. Ideally same for Monolix code

Reviewer #4: Yes

PLOS authors have the option to publish the peer review history of their article (what does this mean?). If published, this will include your full peer review and any attached files.

Reviewer #1: No

Reviewer #2: No

Reviewer #3: No

Reviewer #4: No
---

## [Editor Report · Decision Letter 1]

11 Feb 2021

Dear Mr. Gonçalves,

We are pleased to inform you that your manuscript 'SARS-CoV-2 viral dynamics in non-human primates' has been provisionally accepted for publication in PLOS Computational Biology.

Best regards,

Roland R Regoes

Associate Editor

PLOS Computational Biology

Rob De Boer

Deputy Editor

PLOS Computational Biology

---

## [Editor Report · Acceptance letter]

13 Mar 2021

PCOMPBIOL-D-20-01686R1 

SARS-CoV-2 viral dynamics in non-human primates

Dear Dr Gonçalves,

I am pleased to inform you that your manuscript has been formally accepted for publication in PLOS Computational Biology. Your manuscript is now with our production department and you will be notified of the publication date in due course.

With kind regards,

Alice Ellingham
